

# Pre-competition mental energy and performance relationships among physically disabled table tennis players

Wen-Chuan Chuang[1,4], Frank J.H. Lu[1], Diane L. Gill[2] and Bin-Bin Fang[1,3]

[1] Graduate Institute of Sport Coaching Science, Chinese Culture University, Taipei, Taiwan
[2] Department of Kinesiology, University of North Carolina at Greensboro, Greensboro, NC, USA
[3] School of Physical Education, Quanzhou Normal University, Quanzhou, Fujian, China
[4] General Education Center, National Kaohsiung University of Hospitality and Tourism, Taiwan

## ABSTRACT

Energy is essential to human daily functioning and performance. However, the association of mental energy with athletes' performance has rarely been examined. We attempted to examine the pre-competition mental energy–performance relationships by two studies. Study 1 administered Athletic Mental Energy Scale (AMES, *Lu et al., 2018*) to nine elite physically-disabled table tennis players one day before competition in 5 international tournaments. Then, we collected their subjective performance after each competition. In Study 2, we sampled 77 National-level physically-disabled table tennis players and examined the pre-competition mental energy-performance relationship as the procedure in Study 1. Results from Study 1 provided initial findings of how pre-competition mental energy is associated with performance and portrayed in elite physically-disabled table tennis players. Results from Study 2 further confirmed the pre-competition mental energy- performance relationships. We suggested future studies to examine the mental energy–performance relationships in physically-disabled and abled athletes and different sports.

## INTRODUCTION

Energy is defined as "*the capacity for doing work* (*Giancoli, 2009*, p. 172)". In the physical world, there are many types of energy such as solar energy, thermal energy, nuclear energy, gravitational energy, chemical energy, and hydroelectric energy. Because of so many physical energies they enable machines to produce commercial products or other daily necessities for our life. For human beings, we eat all sorts of food such as fruits, vegetables, meats, grains and many others. The food that we eat then transferring into energy so we can move, function, and engage in daily activities (*Whitney & Rolfes, 2016*). In the psychology domain, researchers believe that mental energy has the same function in our everyday life. Mental energy is defined as "…*an individual's ability to continue long hours of thinking, concentrating attention, and blocking distractions to achieve a given task* (*Lykken, 2005*)".

Corresponding author
Frank J.H. Lu, frankjlu@gmail.com

Despite the importance of mental energy, sport psychologists rarely examine what mental energy is, and how it affects athletes' behavior and performance. Several sport psychologists began to refer to the concept of mental energy and its effects on sport performance in the 1990s. For example, in a narrative description of peak performance in sports, *Loehr (2005)* contended that the management of energy is essential to achieve the best results of sports endeavors. *Loehr (2005)* proposed a pyramid structure of energy which indicated that physical energy is the basic foundation of peak performance. Then, emotional energy is located in the second level which is considered to guide physical energy. Following this conceptual model, mental energy is located in the third level which guides emotional energy and physical energy in the right direction. The top of the pyramid is the spiritual energy which serves as a guide of all types of energy. Specifically, *Loehr (2005)* contended mental energy regulates athletes' high-order functioning such as cognition, perception, abstract thinking, creativity, self-awareness, and regulation. Thus, without mental energy athletes are unable to achieve their best performance.

In the same manner, *Suinn (1986)* published a manual entitled "Seven Steps to Peak Performance", which proposed the psychological skill called visual motor behavior rehearsal (VMBR) to guide athletes' mental energy before the competition. According to *Suinn (1986)*, every athlete has the potential to fulfill his/her potential as long as he/she can guide his/her energy into competition fields, facilities, and sporting tools. By appropriately guiding his/her energy in competition, he/she can enter a state of calm, confidence, and concentration. And, this state is believed to help athletes achieve peak performance.

Thus, it seems that sport psychologists recognize mental energy is important to athletes' behavior and performance. Unfortunately, mental energy received little attention in contemporary sport psychology research. To fill this gap, *Lu et al. (2018)* adopted a theoretical framework of the North American Branch of the International Life Science Institute (ILSI) and proposed a 6-factor Athletic Mental Energy (AME) model. *Lu et al. (2018)* defined athletic mental energy as "…*an athlete's perceived existing state of energy which is characterized by its intensity in motivation, confidence, concentration, and mood*", and developed a measure termed Athletic Mental Energy Scale (AMES) with 6 studies. *Lu et al. (2018)* reported that AMES is a sport-specific measure with sufficient content validity, factorial structure, nomological validity, discriminant validity, and predictive validity. AMES comprises two major components-cognitive factors (*i.e.*, motivation, concentration, and confidence) and emotional factors (*i.e.*, vigor, tirelessness, and calm).

The emotional factors of mental energy can be traced back to the earlier research when researchers used visual analog scales, POMS, and SF-36 Health Survey to measure the emotional factors mental energy (*O'Connor, 2006*). Using emotion to assess mental energy has its' tradition in nutrition studies (*e.g.*, *Chait, 1994*). Similarly, following ILSI's conceptualization of mental energy, researchers also used diverse measures to assess cognitive factors of mental energy including memory tests, electrophysiological indices, brain scanning (*Kennedy et al., 2004*), and attention tests (*Kennedy et al., 2004*; *Lieberman, 2006*). Thus, although mental energy and existing concepts such as mood, emotion, self-efficacy are different constructs, scientists use them to assess mental energy.

Emotional components of vigor, tirelessness, and calm play important roles in sports performance. Conceptually, vigor and tirelessness are the same constructs. It is because *Lu et al. (2018)* used exploratory factor analysis (EFA) to explore the construct of mental energy, and found tireless separated from vigor (*Lu et al., 2018*, p.5). Vigor represents an individual's subjective feeling with heightened arousal (*Lane & Terry, 2000*). In a recent meta-analysis conducted by *Lochbaum et al. (2021)*, it was found that there is a moderate effect size for the vigor-performance relationship (Hedges $g = 0.38$) which supported Morgan's mental health model (*Morgan, 1979*). In line with this consideration, *Campbell & Jones (1994)* examined emotional states in physically disabled athletes and found wheelchair sport participants exhibited an iceberg profile of positive well-being with lower confusion and higher vigor than the sport nonparticipant group. In addition to vigor and tirelessness, past research examined psychological state of elite athletes in peak performance and found athletes frequently reported they experience the sensations of relaxation, calm, effortless, automatic, and non-thinking of performance (*Csikszentmihalyi, 1990*; *Jackson & Roberts, 1992*; *Privette, 1983*; *Ravizz, 1977*). A recent study exploring the optimal psychological state of Australian elite athletes, *Anderson, Hanrahan & Mallett (2014)* found their participants identified calm and relax as the significant experiences.

On the other hand, the cognitive components of athletic mental energy such as motivation, confidence, and concentration have been identified in Olympic and elite athletes (*Fletcher & Sarkar, 2012*; *Gould, Dieffenbach & Moffett, 2002*; *Orlick & Partington, 1988*). For example, in an attempt to understand the role of mental factors in Olympic success, *Orlick & Partington (1988)* investigated 235 Canadian Olympic participants. They found the total commitment to pursuing excellence, mental readiness, attention focus, and some of the other factors that lead to Olympic success for Olympic medalists and world champions. Similarly, to understand the psychological characteristics and talent development in elite athletes, *Gould, Dieffenbach & Moffett (2002)* interviewed 10 U.S Olympic champions, their coaches, and significant others and found the ability to block distraction, confidence, the ability to cope with anxiety as important psychological factors in athletic success. Further, to explore the relationship between psychological resilience and optimal sport performance, *Fletcher & Sarkar (2012)* interviewed 12 Olympic champions about their experiences of withstanding pressure during their sporting careers. Results found motivation, confidence, focus, and some factors protect them from the potential negative effect of stressors. Thus, it seems that athletic mental energy comprises those emotional and cognitive factors that link to sports success.

Recently, researchers used AMES to examine its' influence on athletes' performance and psychological responses. For example, in *Lu et al. (2018)* study, it was found AMES predicted martial art participants' winning the medals. Specifically, among all 6 factors of AMES, two factors of cognitive components- motivation and confidence are the significant predictors of winning the medals; while two factors of emotional components-calm and tirelessness predicted winning medals. Further, in considering that athletic mental energy might be a positive strength that can moderate athletes' life stress-burnout relationship, *Chiou et al. (2020)* conducted two studies to examine moderating effects of athletic mental energy on the life stress-burnout relationship. Both studies found athletic mental energy

moderated athletes' life stress-burnout relationship. Furthermore, in a Turkish study, *Yildiz et al. (2020)* sampled 254 professional football players and administered them with Three-Factor Eating Questionnaire (TFEQ-R18) and 6-factor, 18-item AMES. Results found cognitive restraint eating was positively correlated with athletic mental energy while emotional eating/uncontrolled eating was negatively correlated with athletic mental energy.

Thus, it seems that athletic mental energy might influence athletes' behavior and psychological responses. However, research on athletes' mental energy remains underdeveloped. More empirical studies are needed to confirm whether athletes' mental energy influences athletes' behavior and psychological responses and how mental energy influences athletes' performance. *Martens* (*1987*, p. 49) suggested that if we know little about something in sport psychology, the ideographic approach is an alternative way to gain knowledge. Thus, the purpose of the present study was to sample elite physically disabled table tennis players to examine the pre-competition mental energy-performance relationships. In Study 1, we used both ideographic (within-subject) and cross-sectional (between-subject) designs to find our answers. In Study 2, we extended Study 1 by using a larger sample size to examine the pre-competition mental energy-performance relationship.

Physically disabled athletes are a special population in the sporting world. Although physically disabled athletes have diverse disabilities in trunks/limbs, they engage in diverse competitive sports such as basketball, goalball, rugby, athletics, archery, shooting, powerlifting, and many others (*IPC, 2021*). Research suggests that engaging in competitive sports is beneficial to physically disabled athletes' cognitive function (*Di Russo et al., 2010*), resilience and courage (*Sikorska & Gerc, 2018*), self-esteem (*Van de Vliet, Van Biesen & Vanlandewijck, 2008*), and social function (*Smith, Bundon & Best, 2016*).

Recently, the psychological factors for elite physically disabled athletes received much attention from researchers. For example, *Martin (2016)* indicated that to achieve peak performance, physically disabled athletes need strong motivation to engage in training and competition. In addition, they need to regularly engage in psychological skills training to cope with stress and anxiety in competition. The other researchers (*e.g.*, *Dieffenbach & Statler, 2012*) contended that for top disabled athletes their needs and experiences of psychological skill training and services are more similar to able-bodied elite athletes than different. To prepare disabled athletes to participate in the 2014 Paralympic winter games, *Martin (2012)* suggested that disabled athletes must work intensively with sport psychologists. They need to learn coping skills, practice imagery, boost confidence, use positive self-talk, and regulate positive emotion before the competition. Thus, it seems that psychological factors for elite physically disabled athletes in competitive sports are similar to those abled athletes. However, how these psychological factors influence physically disabled table tennis players in the competition has never been examined. Especially, will the emotional and cognitive factors of athletic mental energy associated with their performance be the main purpose of the present study. Extending this line of research, the purpose of the present study was to examine how mental energy is associated with physically-disabled table tennis players' competition performance.

## METHODS

### Study 1

#### Purpose

The purpose of Study 1 was to examine the pre-competition mental energy-performance relationship in elite physical physically disabled tennis players. We adopted *Lane & Chappell*'s *(2001)* approach by using both ideographic (within-subject) and cross-sectional (between-subject) designs to achieve this purpose.

#### Methods

*Participants.* Nine elite disable table tennis players ($n = 9$, *M* age $= 42.11$ years, SD $= \pm19.02$ years; Males=4, Females=5) who trained 4.12 mean hours (SD $= 1.85$) per-day and 20.10 h per-week (SD $= 10.41$) were invited as participants. Participants had been playing an average of 23.67 years (SD $= \pm7.37$ years) and had won many international tournament titles (ranging from 1 to 14 titles.) During data collection, they were participating in the 2019~2020 Para Table Tennis (PTT) scheduled international tournaments.

*Procedures.* Before data collection, the researchers gained approval from the Antai-Tian-Sheng Memorial Hospital Institutional Review Board (TSMH IRB No:20-123-B). Then, the first author contacted the target team's coaches through emails and phones and briefly informed them of the purpose of the research. After their agreement, we made an appointment to meet participants and informed them that they were part of a research program to examine their competition experiences. Also, the collected data would serve as a reference for making team's training schedule in the future. To promote honesty, participants were informed that there are no right or wrong answers for the questionnaire responses, and all players' answers would be treated with strict confidentiality. After the agreement, the players signed the consent forms and participated in a drawing to pick an assigned number to serve as an anonymous ID for the follow-up investigations. Because mental energy is a state construct, it is suggested that it should be measured near competition to reflect exact psychological state (*e.g.*, *Campbell & Jones, 1997*; *Hanton, Thomas & Maynard, 2004*). However, the head coach of the National team told us that during the day of competition the schedule is tight. Players have no mood to complete the test carefully. After a negotiation, participants completed the Athletic Mental Energy Scale (AMES, *Lu et al., 2018*) one day before the competition and completed the self-referenced performance measure approximately one hour after the competition.

#### Measures

*Athletic Mental Energy Scale (AMES).* The 18-item AMES (*Lu et al., 2018*) was used to assess participants' perception of their existing energy state. The 18-item AMES has 6 factors, and three items in each factor as follows: (a) vigor (sample question as: "I feel there is an endless energy coming from my body"), (b) confidence (sample question as: "I feel I can win all competitions in the future"), (c) motivation (sample question as: "I want to show my best to others in sports"), (d) tireless (sample question as: "No matter how long the training lasts I don't feel tired"), (e) concentration (sample question as: " I am free of distraction during competition and training"), and (f) calm (sample question as: "Facing

to coming competitions I don't feel anxious"). When answering AMES, participants have to identify the feeling of each item on a 6-point Likert scale that ranged from 1 (not at all) to 6 (completely so). The Cronbach's $\alpha$ of six factors of the AMES in this study ranged from .80 to .92.

*Measures of performance.* When participating in an international tournament, each player faces diverse opponents with different abilities. Also, there were different competition schedules in the tournaments. For example, some players might participate in a single competition while others might participate in a group competition. Further, some players might participate in both single and group competitions. It is very difficult to use an objective performance score due to such diverse conditions. Therefore, we adopted previous researchers' suggestions (*e.g.*, *Beedie, Terry & Lane, 2000*; *Lane & Chappell, 2001*; *Prapavessis, 2000*) by using a self-referenced measure. In this study, each participant rated their performance by responding following question: "How do you feel about your performance in the last game?" Based on their perceptions they rated a score ranging from 0 to100 points in a response sheet.

### Statistical analyses

We analyzed pre-competition mental energy and performance relationships via descriptive statistics and Pearson Correlation for each individual across the 5 international tournaments. The descriptive statistics allow us to understand nine players' mental energy states in five games, and how these states are associated with their performance. Further, we used multiple regression to examine which mental energy factors predicted performance.

*Results.* Because each factor has three items to respond to, we used a 6-point Likert scale, so the highest score in AMES is 18 and the lowest score is 3, and the median score is 9 for each factor. As Table 1 indicates player #1 has the highest mental energy score in each factor, and player #6 has the lowest mental energy score. Among all players, 6 players scored higher above 9, while 3 scored equal or lower than 9. At Game #1, they reported high motivation and concentration but low tireless. At Game #2, all mental energy factors dropped. At Game #3, most factors of mental energy were similar to Game #2, but they reported low tireless. At Game #4 and Game #5, the mental energy did not change much, still, they scored low in tireless. Further, across 5 international tournaments, participants reported their vigor, confidence, motivation, concentration, and calm around 11 points, and tireless lower than 10. Table 2 shows the pre-competition mental energy-performance relationship. A significant relationship was found between vigor and performance in player #1 and player #2 and tireless performance in player #2. Most factors in each player remained insignificant. Further, we combined all participants' data and examined the pre-competition mental energy-performance relationship. As Table 3 shows, in Game #1, there was a significant relationship between tireless and performance. Also, in Game #4, there is a significant relationship between vigor-, motivation-, concentration-, and calm-performance relationship. In other games, most mental energy-performance relationships were insignificant. Moreover, a multiple regression found total mental energy accounted for

**Table 1 Descriptive statistics for pre-competition AMES.** This table describes each participant's mental energy across five games.

| Player/Game | Vigor | | Confidence | | Motivation | | Concentration | | Tireless | | Calm | |
|---|---|---|---|---|---|---|---|---|---|---|---|---|
| | M | SD | M | SD | M | SD | M | SD | M | D | M | SD |
| Player 1 | 14.7 | 1.4 | 13.3 | 1.4 | 15.7 | 1.9 | 15.0 | 0.9 | 12.5 | 0.5 | 14.3 | 0.8 |
| Player 2 | 11.8 | 0.4 | 11.2 | 1.6 | 11.6 | 0.5 | 10.6 | 1.5 | 11.0 | 1.2 | 11.6 | 0.9 |
| Player 3 | 13.7 | 2.3 | 13.5 | 2.1 | 14.5 | 2.3 | 14.8 | 1.9 | 14.2 | 2.3 | 12.5 | 1.8 |
| Player 4 | 13.0 | 2.0 | 11.8 | 0.4 | 12.7 | 1.6 | 12.8 | 2.0 | 12.0 | 0.0 | 12.2 | 0.4 |
| Player 5 | 9.0 | 2.5 | 10.8 | 1.6 | 10.8 | 2.4 | 11.0 | 3.3 | 9.4 | 2.6 | 11.0 | 3.5 |
| Player 6 | 8.3 | 0.8 | 9.5 | 1.4 | 9.0 | 0.6 | 9.3 | 1.2 | 7.0 | 1.3 | 7.8 | 1.0 |
| Player 7 | 8.5 | 1.4 | 10.2 | 1.8 | 10.2 | 1.6 | 7.7 | 2.1 | 5.7 | 2.3 | 10.3 | 2.7 |
| Player 8 | 10.2 | 1.6 | 10.0 | 0.7 | 12.8 | 1.3 | 13.4 | 2.2 | 7.4 | 1.1 | 12.8 | 0.8 |
| Player 9 | 11.0 | 0.0 | 10.6 | 0.9 | 9.4 | 0.5 | 9.8 | 0.8 | 9.0 | 0.0 | 10.0 | 0.7 |
| Game 1 | 12.8 | 2.8 | 12.0 | 3.0 | 13.3 | 3.3 | 13.1 | 4.2 | 11.1 | 3.6 | 12.6 | 3.5 |
| Game 2 | 11.0 | 1.4 | 10.9 | 1.2 | 11.2 | 2.0 | 11.6 | 1.9 | 10.3 | 1.8 | 11.2 | 2.1 |
| Game 3 | 10.4 | 2.7 | 10.8 | 1.6 | 11.4 | 2.2 | 10.8 | 2.8 | 9.3 | 2.8 | 11.0 | 1.7 |
| Game 4 | 11.1 | 3.3 | 11.3 | 2.0 | 11.4 | 2.7 | 10.8 | 3.0 | 9.4 | 3.8 | 10.8 | 2.4 |
| Game 5 | 10.4 | 2.5 | 11.0 | 1.6 | 11.6 | 2.2 | 11.4 | 2.9 | 9.4 | 3.7 | 10.8 | 2.3 |
| All Game | 11.2 | 2.6 | 11.2 | 1.9 | 11.9 | 2.9 | 11.5 | 3.0 | 9.9 | 3.1 | 11.2 | 2.4 |

**Table 2 Idiographic analysis of the relationships between pre-competition AMES and performance.** This table describes mental energy and performance relationship among nine players.

| Player | Vigor-performance | Confidence-performance | Motivation-performance | Concentration-performance | Tireless-performance | Calm-performance |
|---|---|---|---|---|---|---|
| 1 | 0.89[*] | 0.00 | −0.41 | −0.53 | 0.61 | −0.13 |
| 2 | 0.88[*] | 0.10 | −0.10 | −.022 | 0.91[*] | 0.25 |
| 3 | 0.41 | 0.43 | 0.40 | 0.23 | 0.00 | 0.13 |
| 4 | 0.39 | 0.32 | 0.32 | 0.32 | 0.15 | 0.32 |
| 5 | 0.00 | −0.27 | 0.28 | 0.17 | −0.30 | 0.00 |
| 6 | 0.38 | −0.09 | −0.79 | 0.34 | 0.77 | −0.38 |
| 7 | 0.12 | 0.32 | −0.16 | 0.30 | 0.45 | −0.03 |
| 8 | 0.30 | 0.00 | 0.56 | 0.17 | 0.32 | 0.33 |
| 9 | – | −0.52 | −0.33 | 0.17 | – | 0.82 |

**Notes.**
[*] $P < .05$.
[**] $P < .10$.

12% of the variance in performance (Adj $R^2 = .12$, $p < .05$), with vigor (Beta $= .87$, $p < .05$), and tireless (Beta $= −1.0$, $p < .01$) as the two significant predictors of performance.

*Conclusion.* The results of Study 1 provided initial but valuable findings. By ideographic (*i.e.*, within-subject) analyses, it was found each elite physically disabled tennis player showed different levels of mental energy before the competition. Further, the associations between pre-competition mental energy and performance were personal-specific and individualized. We found some players exhibited a significant pre-competition mental energy-performance relationship while other players remained insignificant. Further,

**Table 3 Cross-sectional analysis of AMES and performance relationships in five games.** This table illustrates how precompetition mental energy correlates performance in five games.

| Game | Vigor-performance | Confidence-performance | Motivation-performance | Concentration-performance | Tireless-performance | Calm-performance |
|------|-------------------|------------------------|------------------------|---------------------------|----------------------|------------------|
| 1 | −0.46 | −0.27 | −0.20 | −0.24 | 0.53 | −0.44 |
| 2 | 0.00 | −0.56 | −0.04 | −0.09 | 0.63[*] | −.040 |
| 3 | −0.05 | 0.00 | −0.26 | −0.16 | −0.31 | 0.42 |
| 4 | 0.79[*] | 0.56 | 0.79[*] | 0.75[*] | 0.51 | 0.72[*] |
| 5 | −0.31 | −0.46 | −0.43 | −0.52 | −0.48 | −0.40 |

Notes.
[*]$P < .05$.
[**]$P < .10$.

the cross-sectional (*i.e.*, between-subject) analyses found there were some significant associations between pre-competition mental energy and performance. However, due to the small sample size, the results were not overall significant. Thus, using a larger sample size to examine whether pre-competition mental energy is associated with performance is needed. Therefore, we conducted Study 2 to find our answers.

### Study 2

*Purpose.* The purpose of Study 2 was to replicate Study 1 and provide more evidence by using a larger sample size to examine the pre-competition mental energy-performance relationship. Specifically, we hypothesized that all subcomponents of the mental energy (*i.e.*, vigor, confidence, motivation, concentration, tireless, and calm) will associate and predict physically diabled table tennis players' performance.

### Methods

*Participants.* Before data collection, we used G ∗Power 3.1.7 to determine our sample size. First, we chose ''Exact'' for the test family with a bivariate normal model of correlation. Then, in the input parameter, we chose a two-tailed test with Effect Size = .50; $\alpha$ = .05, and power = .95, it was suggested that sample size was 46 would be appropriate. Thus, in a domestic national-level championship we recruited 77 physically disabled tennis players as our participants (Males = 53; Females = 24) with a mean age of 47.67 years (SD = ±14.36). Participants had been playing an average of 16.31 years (SD = ±20.68 years). The criteria for selecting participants are as follows: (a) engaging in disabled table tennis competitive sports; (b) engaging in regular training daily and weekly; (c) having at least one time in a disabled table tennis competition. The exclusion for participating in this study is as follows: (a) recreational physically disabled table tennis participants without regular and formal training; (b) inexperienced disabled table tennis athletes who have no competition experiences before. By such selection criteria, we hope to recruit participants with full experience in sports training and competition.

*Measurements and procedures.* After receiving approval from Antai-Tian-Sheng Memorial Hospital Institutional Review Board (TSMH IRB No:20-123-B), we contacted each team's head coach and informed them of the purposes and procedures of our study. We met
**Table 4  Descriptive statistics and bivariate correlations of the study variables.** This table uses a crossectional approach to explore precompetition mental energy and performance relationship.

|  | 1 | 2 | 3 | 4 | 5 | 6 | 7 |
|---|---|---|---|---|---|---|---|
| 1 Vigor | 1 | | | | | | |
| 2 Confidence | 0.66** | 1 | | | | | |
| 3 Motivation | 0.78** | 0.66** | 1 | | | | |
| 4Concentration | 0.70** | 0.61** | 0.70** | 1 | | | |
| 5 Tireless | 0.79** | 0.65** | 0.61** | 0.66** | 1 | | |
| 6 Calm | 0.55** | 0.73** | 0.53** | 0.51** | 0.49** | 1 | |
| 7 Performance | 0.25* | 0.37** | 0.28* | 0.25* | 0.23* | 0.29* | 1 |
| Mean | 13.31 | 12.44 | 13.31 | 13.26 | 11.45 | 12.43 | 68.70 |
| SD | 2.90 | 2.55 | 3.04 | 2.97 | 3.34 | 2.85 | 17.72 |
| $\alpha$ | 0.90 | 0.74 | 0.84 | 0.83 | 0.90 | 0.82 | |

**Notes.**
*$P < .05$.
**$P < .10$.

each team one night before the competition and informed them of the general purposes and procedures of our study. We informed them the study was to explore participants' experiences in physically-disabled table tennis competitions. There were no right or wrong answers for their responses in questionnaires. All collected data would remain anonymous, confidential, and used for group analyses. After they agreed to participate, they signed a consent form and completed the demographic questionnaire and the Athletic Mental Energy Scale (AMES, *Lu et al., 2018*). The demographic questionnaire and AMES were similar to Study 1. However, in Study 2, the demographic questionnaire has an item that requests participants to fill four digits of their IDs so we can match their performance the next day. We collected their self-referenced performance measures the next day approximately one hour after the competition.

*Statistical analyses.*  In Study 2, we used Pearson Product Moment Correlation (PPMC) to examine the pre-competition mental energy-performance relationship. Also, to understand what factor of mental energy predicted the performance, we used hierarchical regression to examine which mental energy variable predicted performance after controlling gender, age, training hours per week, and years of sport experiences (*Aljandali, 2017*).

*Results.*  As Table 4 illustrates, all subscales of mental energy showed appropriate internal reliability ($\alpha = 0.74$–0.90). Also, participants had high scores in vigor, motivation, and concentration, but low scores in tireless. The lower score in tireless is similar to Study 1. Generally, participants in Study 2 had higher scores in all mental energy than Study 1. Further, it was found all mental energy factors had high correlations with each other. All mental energy factors significantly correlated with performance ($r = .23 \sim .37$). Further, after controlling gender, age, training hours per week, and years of sport experiences, confidence was the only predictor of all mental energy factors which explained 12% of the variance (Adj $R_2 = .12$, $p < .05$; Table 5).

**Table 5  Prediction of athletic mental energy on performance (Study #2).** This table illustrates the hierarchical regression of the prediction of athletic mental energy on performance.

| Model | R | R-squr. | Adjt. R-squr. | R-squr. Change | F Change | Sig. F Change | Unstad. Coeffic. | Stand. Coeffic. |
|---|---|---|---|---|---|---|---|---|
| 1 | .22 | .05[*] | .04 | .05 | 3.85 | .05 | −8.40[*] | −.22[*] |
| 2 | .24 | .06 | .03 | .01 | .70 | .41 | .12 | .10 |
| 3 | .26 | .07 | .03 | .01 | .55 | .46 | −.79 | −.09 |
| 4 | .41 | .17[**] | .12 | .10 | 8.94 | .00 | 2.34[**] | .34[**] |

**Notes.**
[*]$p < .05$
[**]$p < .01$
[a]Predictors in the Model: (Constant), gender.
[b]Predictors in the Model: (Constant), gender, age.
[c]Predictors in the Model: (Constant), gender, age, dailytrain.
[d]Predictors in the Model: (Constant), gender, age, dailytrain, confidence.
[e]Dependent Variable: subjective performance.

*Conclusion.* Compared to Study 1, Study 2 provides more information about the pre-competition mental energy-performance relationship is physically disabled table tennis players. First, we found AMES has appropriate internal reliability, and subscales have high correlations with each other as *Lu et al. (2018)*. Further, compared to participants in Study 1, the participants in Study 2 had higher mental energy. Furthermore, all participants in Study 1 and Study 2 showed low score in tireless. Moreover, we found there is a significant pre-competition mental energy-performance relationships, and confidence was the strongest predictor of performance in Study 2.

# DISCUSSION

## Theoretical contributions/implications

In considering that mental energy is important for human daily functioning and performance, this study examined the pre-competition mental energy-performance relationship in physically disabled table tennis players. The Study 1 found elite physically-disabled tennis players showed different levels of mental energy before the competition. Also, we found that the associations between pre-competition mental energy and performance were personal-specific and individualized. Study 2 found there is a significant pre-competition mental energy-performance relationship. Further, we found confidence as the strongest predictor of performance in Study 2. The results have several implications for researchers.

First, this study provides initial knowledge of pre-competition mental energy and performance relationships in sports. Despite sport psychologists proposing the effects of mental energy on athletes' performance in the 1990s, related empirical reports are still scarce. Thus, our study fills a gap after *Lu et al. (2018)* study.

The emotional components of mental energy (*i.e.*, vigor, tireless, and calm of the AMES) and their relationships with performance are novel and filled with implications. For example, the vigor-performance association reflects an early study on the iceberg profile in which *Morgan (1979)* found successful Olympic rowing athletes were high in vigor but low in fatigue, depression, anger, confusion, and anxiety. Vigor is defined as

an individual's subjective feeling with heightened arousal (*Lane & Terry, 2000*). Thus, with heightening vigor, it is reasonable that an athlete would maximize his/her efforts in enhancing performance. Further, we found there is a significant relationship between tireless and performance. Theoretically, this relationship is similar to the vigor-performance relationship because *Lu et al. (2018)* contended that tirelessness was derived from vigor through factor analyses.

The calm-performance relationship is meaningful in sport psychology because research found that when in a peak performance athletes experience a state of calm such as "no fear of failure" and "physically and mentally relaxed" (*Loehr, 1984*). Generally, when entering competition athletes increase their performance anxiety (*Cheng, Hardy & Markland, 2009*). The increased anxiety may lead to over-arousal and distraction and eventually lead to choking (*Baumeister & Showers, 1986*). If an athlete can remain calm during the competition, he/she would be able to perform well.

The cognitive components of mental energy (*i.e.*, concentration, confidence, and motivation of the AMES), and their relationships with performance are also insightful. In competition settings, it is important to maintain concentration to perform well (*Williams et al., 2015*). Concentration is defined as one's cognitive ability to block distractions and focus one's attention on a given task (*Weinberg & Gould, 2015*). It is also found that when athletes achieve their peak performance they were "*able to focus tasks at hand*" and "*emerge in the activity that they engage* (*Williams et al., 2015*)". Research has found that athletes who scored high in concentration performed better (*Paul & Garg, 2012*; *Paul, Garg & Sandhu, 2012*). Thus, the positive concentration-performance relationship in our study is consistent with previous research.

Further, we found a significant relationship between confidence and performance, and confidence was the strongest predictor of performance. Confidence is an influence on athletes' thoughts, feelings, and behavior (*Vealey & Chase, 2008*). Most renowned athletes (*e.g.*, Mohammed Ali, Serena Williams, & Tiger Woods) are famous for their self-assurance in sports. *Vealey & Chase (2008)* contended that heightened confidence enhances athletes' positive emotions, efforts, and persistence. Therefore, a high confidence athlete can achieve better performance than those with lower confidence. Thus, the significant confidence-performance relationship in our study echoed *Vealey & Chase (2008)* sport-confidence model. Why physically-disabled tennis players perceived confidence as the strongest predictor of their performance is unknown. *Vealey & Chase (2008)* contended that athletes' characteristics and organizational culture affect the manifestation of sport confidence. Thus, future study is needed to further confirm whether physically disabled tennis players' characteristics and culture affect their confidence and subsequent performance.

The significant motivation-performance relationship highlights the role of motivation in sports. Motivation refers to the "direction and intensity of behavior (*Gill, Williams & Reifstech, 2017*). Motivation determines how much effort one exerts on a given task. With high motivation, an athlete would spend more time training and persist longer when facing adversity. Research indicates that disabled athletes have diverse motives to participate in competitive sports (*Mumcu et al., 2017*). Also, some researchers found participating in competitive sports provided a means of affirming competence, promoting fitness, and

delaying disabilities (*Page, O'Connor & Petersen, 2001*). The physically-disabled table tennis players in our study have overcome physical disabilities and engaged in sports training and competitions, and their motivation is reflected in the mental energy associated with performance.

Thus, our study extends past research in the psychology of disabled athletes (*e.g.*, *Dieffenbach & Statler, 2012*; *Martin, 2012*; *Martin, 2016*) and found psychological factors such as vigor, calm, confidence, motivation, and focus not only manifested in athletic mental energy but also associated with competition performance. The results echoed research in the psychology of sport excellence (*e.g.*, *Fletcher & Sarkar, 2012*; *Gould, Dieffenbach & Moffett, 2002*; *Orlick & Partington, 1988*) that psychological factors play important roles in athletic success. Further, our study might prompt researchers and practitioner to engage in the service and study of the psychology of the disabled athletes.

Despite these initial findings, it needs to discuss our results in some way. First, it is the mechanism that underlies the athletic mental energy-performance relationship. Athletic mental energy is defined as an athlete's perceived existing state of energy which is characterized by its intensity in motivation, confidence, concentration, and mood (*Lu et al., 2018*)''. It is plausible that such perception makes athletes feel strong, calm, and concentrated; which in turn boosts their performance. Despite this speculation, more research is needed to examine the triangular relationships among mental energy, psychological states, and performance. Further, because the concept of athletic mental energy is still under development, does athletic mental energy is manifested only by 6 psychological factors such as confidence, motivation, concentration, vigor, tirelessness, and calm needs further examined. Furthermore, it is the strength of the athletic mental energy-performance relationship. We found all correlations ranged from .23 to .37 with a moderate magnitude which means that this relation is not strong (*Cohen, 2013*). So far, there are only a few studies to examine athletic mental energy relationships. Despite our study supports *Lu et al. (2018)* study, due to the differences in competition nature, contexts, skill levels, and participants, it is difficult to generalize our results to other sports such as soccer, track and field, weight-lifting, and others. Future study is needed to examine the prediction of athletic mental energy on competition performance in different sports.

## Strengths of the study

There are several strengths of our study. First, we adopted an empirical approach to examine mental energy-performance relationships by two studies with different methods. *Brustad (2002)* suggested that using multi-method approaches to address any research question allows researchers to take a broader perspective than with a one-shot study. In Study 1 we adopted *Martens (1987)* suggestion by applying an ideographic approach to invite nine elite physically disabled table tennis players as participants. Although this was a small sample, Study 1 portrayed what elite physically disabled table tennis players feel about mental energy and how mental energy is associated with their performance across five international tournaments. Study 2 is nomothetic in nature. It allows researchers to understand how pre-competition mental energy correlates with physically-disabled tennis players' competition performance, and which mental energy variables predict performance.

Further, both studies adopted a longitudinal approach and collected data from the same sample over an extended period of time. By such design, we can evaluate the relationship between pre-competition mental energy and performance over time (*Caruana et al., 2015*).

**Limitations and future suggestions**

Despite these initial findings, several limitations should be noted. First, due to the limited number of physically disabled individuals participating in table-tennis, we are unable to recruit a larger sample for our study. We used a conservative way to estimate our sample size (*i.e.*, effect size $= .50$; $\alpha = .05$, and power $= .95$). We suggest future studies may recruit a larger sample in other sports or abled athletes in order to find better evidence. Second, to avoid disturbance before the competition, we did not measure participants' mental energy very close to the competition (*e.g.*, 30 min before the competition). Whether our measurement of mental energy reflects participants' existing state of mental energy at that moment needs further examination. Past research on pre-competition anxiety patterns generally measured participants' responses closer to competition (*e.g.*, *Campbell & Jones, 1997*; *Hanton, Thomas & Maynard, 2004*). We suggest future research measure athletes' mental energy as close to competition as possible. Further, we did not collect abled athletes' mental energy and performance at the same time. Also, whether the pre-competition mental energy-performance relationships of our study are similar to those for abled athletes needs further examination. Further, we used table tennis players as our participants whether our results can be generalized into other strenuous sports such as weight-lifting, 100-meters dash, shot-put, and hammer-throwing, future study is needed because fine motor skills and gross motor skill sports have different requirements for the physical energy expenditure and muscles involved (*Schmidt et al., 2019*). Moreover, because our study is observational research we cannot determine cause-and-effect. We suggest future studies examine whether a mental energy intervention improves athletes' performance. Furthermore, we adopted *Lu et al. (2018)* athletic mental energy as our working definition. However, *Lu et al. (2018)* definition warrant greater focus because they used the consequences of energy are part of the definition. If someone is displaying intense confidence does this, therefore, mean that they are displaying intense energy? Therefore, future study is needed to examine the definition and conceptualization of athletic mental energy.

# CONCLUSION

The athletic mental energy is a newly emerging concept in sport psychology. However, very few studies examined this mental energy-performance relationship in sports settings. We used two studies to examine the mental energy-performance relationships among physically-disabled table tennis players and provided preliminary results. We suggested future studies may examine whether mental energy is correlated with competition performance both in abled and disabled athletes and in different sports.

### Funding

This is a self-initiatedd project. No funding form related institute.

### Competing Interests

Frank J. Lu is an Academic Editor for PeerJ.

### Author Contributions

- Wen-Chuan Chuang conceived and designed the experiments, performed the experiments, analyzed the data, prepared figures and/or tables, authored or reviewed drafts of the paper, and approved the final draft.
- Frank J.H. Lu conceived and designed the experiments, analyzed the data, authored or reviewed drafts of the paper, and approved the final draft.
- Diane L. Gill conceived and designed the experiments, prepared figures and/or tables, authored or reviewed drafts of the paper, and approved the final draft.
- Bin-Bin Fang performed the experiments, analyzed the data, prepared figures and/or tables, help arrange participants' competition schedule, and approved the final draft.

### Human Ethics

The following information was supplied relating to ethical approvals (*i.e.*, approving body and any reference numbers):

Antai- Tian-Sheng Memorial Hospital Institutional Review Board

### Data Availability

The raw data are available in the Supplemental Files.

### Supplemental Information

Supplemental information for this article can be found online at http://dx.doi.org/10.7717/peerj.13294#supplemental-information.

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
