# Peer review of "Pre-competition mental energy and performance relationships among physically disabled table tennis players"

_PeerJ, doi:10.7717/peerj.13294_

## Round 0.1 · original submission · Major Revisions

I now have received two reviewers' comments. Although both reviewers expressed their interest in your study, several aspects of this manuscript should be revised to improve its clarity. Their observations are presented with clarity so I'll not risk confusing matters by belaboring or reiterating their comments. While I might quibble with the occasional point, I note that I regard the reviewers' opinions as substantive and well-informed. I believe that all of the highlighted reservations require contemplation and appropriate attention in revising the document if it is to contribute appropriately to Peerj and the extant literature. Please revise or refute according to the two reviewers' comments and provide a point by point reply in addition to the revised manuscript.

Tsung-Min Hung, PhD., FNAK, FISSP
PeerJ Academic Editor
Research chair professor,
Department of Physical Education and Sport Science,
National Taiwan Normal University
Taiwan

·

Basic reporting

An interesting article that was generally well written that investigated relationships between the concept of energy and performance.

The work has value to the sports psychology community and the concept of energy as the authors indicate is one that has not been researched a great deal. However, I would like the authors to expand the literature and provide greater conceptual clarity for how energy links with existing concepts such as mood, emotion, self efficacy etc. It appears they are introducing a relatively new concept and in doing so it would help if it's was conceptualised within the existing literature with greater clarity.

Experimental design

The article is original and within the scope of the journal.

The research question is suitably well described. The authors have provided some indication of how research identifies a gap in the literature.

The methods the authors have a strong degree of rigour. However, I would like to see stronger justifications on when the measures were taken. For example why take measures of energy a day before competition rather than relatively close to performing?

The methods are suitably well defined and described to allow replication.

Validity of the findings

The authors could provide a stronger rationale for the data analysis methods chosen.

At present, they emphasise the significance of the results rather than the strength of relationships.

I would like to see some thought as to the mechanisms underlying hypothesise relationships, that is we know that energy links to performance but why should this be the case? with the subcomponents of the energy scale it might be possible to set an test hypotheses.

Additional comments

Definition of energy warrants greater focus. at present the consequences of energy are part of the definition. If someone is displaying intense concentration does this therefore mean that they are displaying intense energy?
This is debatable because it may be relatively easy for someone to concentrate and therefore intense energy is not necessary. This is just an example of where there is a definition where the consequences other constructs are not distinguish from the construct itself.

Martens (1987, p.49) suggested that if we know little about something in sport
83 psychology, the ideographic approach is an alternative way to gain knowledge. It would be useful to explain why this might be the case

Athletic Mental Energy Scale (AMES, Lu et al., 2018) one day before the competition and completed the self-referenced performance measure approximately one hour after the competition.

Authors should provide a justification for the temporal decisions in their study -for example why one day before competition? This is especially important as emotions and moods can vary from moment to moment and therefore they may have little bearing on mood states going into competition.

Reviewer 2 ·

Basic reporting

Many thanks for reviewing this manuscript Pre-competition mental energy and performance relationships among physically disabled table tennis players. The authors presented a test of the relationship between Pre-competition mental energy and athletes’ performance, by two studies from physically disabled table tennis players. The topic and research questions are novel and significant; however, I have several comments and suggestions that would need to be addressed before I can recommend publication. These are outlined below.

1. Insufficient review and discussion of relevant literature. While authors briefly introduced a relatively novel concept named mental energy in their study, they seemed not aware of much of the mental energy and its components-related literature and theories in psychology, sport, and performance domains. For example, as vigor, confidence, emotions, and other components can be regarded as psychological resources that have been linked to performance, using resource-based theoretical perspectives in sports and performance literature for developing hypotheses and justifying the psychological mechanism of mental energy and performance would provide more solid foundation and insights into their work. Specifically, I found the hypotheses development and the arguments mainly built on reporting past results instead of theoretical justifications, which is, to some degree, insufficient and not convincing. Here I suggest authors draw on literature/theory in sport psychology to provide a more comprehensive and complete theoretical reasoning for the relationship between mental energy and performance.

2. A lack of solid reasoning for the chosen subject. The authors have provided some good reasons for the important target of physically disabled table tennis players and why we should put more attention on them. However, the current reasoning mainly explains the background and current situation regarding participation in competitive sports as well as the benefits of engaging in competitive sports. It is advisable to provide more concise reasons to justify why physically disabled table tennis players would be the important target when investigating mental energy.

Experimental design

3. Methodological concerns. Using two studies to test research hypotheses indeed provides more rigorous results. However, the sample size, in general, is too small to conduct statistical analysis, especially Pearson correlation and multiple regression. Although the reported results were significant as the authors’ expectation, these findings might be relatively unstable and unreliable, as a small sample size decreases the power of the study and enhance the margin of error. Moreover, the current statistical analysis was based on between-person analysis but not within-person analysis according to the analytic methods authors used. To further confirm the nested data structure of study 1, multilevel linear modeling would be necessary to adopt in the study; however, due to the small sample size, HLM is not suitable to use. Here I suggest the authors (1) increase the sample size, (2) pool the sample of study 1 and study 2, or (3) remove study 1.2. A lack of solid reasoning for the chosen subject. The authors have provided some good reasons for the important target of physically disabled table tennis players and why we should put more attention on them. However, the current reasoning mainly explains the background and current situation regarding participation in competitive sports as well as the benefits of engaging in competitive sports. It is advisable to provide more concise reasons to justify why physically disabled table tennis players would be the important target when investigating mental energy.

Validity of the findings

4. Discussions here are more limited around the current results. It would be beneficial to have more consideration, conversation, and discussion of theoretical contribution (or at least broadening the discussion section) in the sport psychology literature (refer to the first point of comment).

Additional comments

Minor Comments
(1) L.110: It is advisable to report more complete demographic information (e.g., gender, weekly training hours…).
(2) L.217-220: Was the measure of performance collected by both subjective and objective data? If so, the authors might analyze both two indices to obtain more rigorous results.
(3) When the AMES was collected? Is it corresponding to the definition of pre-competition? The authors should clearly describe, define, and explain.
(4) It is advisable to add a table to display the regression results.
(5) L.232-233: “only confidence was selected into the model” Did authors actually use hierarchical regression? It seems that step-wise regression was employed but not hierarchical regression. However, using hierarchical regression would be more appropriate in explaining the theoretical implication, so I suggest authors reconfirm their data analytic approach.
(6) L.260-263: “Research in other domains such as nutrition science 
suggested that intake of energetic supplements (e.g. Ginkgo biloba) improves mood and attention 
in healthy subjects, and consuming omega-3 polyunsaturated fatty acids reduced the risk of 
age-related cognitive decline (Gorby, Brownawell, & Falk, 2010).” This sentence seems irrelevant to this study and would be better removed or replace with other arguments.
(7) L.329: It is questionable that the authors mentioned “both studies adopted a longitudinal approach” as study 2 did not use repeated measures across time (i.e., study 2 adopted a time-lagged design).
(8) L.345: Why authors merely recommend future research to focus on strenuous sports? Is there any reason that researchers should put more attention to this type of sport? What about team sports? It would be better to explain more about this idea.

---

## Round 0.2 · Minor Revisions

I have now received the reviewer's comment with a general positive reaction to your reply and revisions from previous comments. However, a few minor issues remained to be addressed before I can accept your manuscript. Please take care of these issues and provide a point by point reply in addition to the revised manuscript.

Tsung-Min Hung, PhD., FNAK, FISSP
PeerJ Academic Editor
Research chair professor,
Department of Physical Education and Sport Science,
National Taiwan Normal University
Taiwan

Reviewer 2 ·

Basic reporting

This revision has improved the paper, theoretically and empirically. I appreciated that attention to detail. I advocate there are some further refinements needed.

Experimental design

The contents and explanations are appropriate.

Validity of the findings

L.324-325: with respect to the use of hierarchical regression analysis, I suggest the sentence “only confidence was selected into the model…” should be revised because all the predictors were in the model and just one predictor was significant. Moreover, for more comprehensive information to readers, it is advisable to report all the values of unstandardized or standardized coefficients of all predictors in Table 5.

Additional comments

The notes of significance in Table 2 to Table 5 need to be corrected. It is “*p < .05, **p < .01” but not “*P>.05,**P>.10”.

---

## Round 0.3 · accepted · Accept

I have read through your revised manuscript and found that you've satisfactorily addressed the few minor issues raised by the reviewer. I think there is no need to send out for further review and decide to accept your manuscript at its current version. You and your coauthors have my congratulations. Thank you for choosing PeerJ as a venue for publishing your research work and I look forward to receiving more of your work in the future.

Tsung-Min Hung, PhD., FNAK, FISSP
PeerJ Academic Editor
Research chair professor,
Department of Physical Education and Sport Science,
National Taiwan Normal University
Taiwan